Connect attack in IoT-WSN detect through cyclic analysis based on forward and backward elimination

http://orcid.org/0000-0003-2708-6337 S. Tamil Selvi ts1171@srmist.edu.in
P. Visalakshi
Networking and Communications, SRM Institute of Science and Technology , Kattankulathur, Tamilnadu , India
Zhang Yue
Electronic publication date: 2024 Jun 28
Publication date: 2024
Volume: 10
Electronic Location ID: e2130
Received 2023 Nov 20; Accepted 2024 May 23
Copyright: © 2024 S and P
Copyright year: 2024
Copyright holder: S and P
License: This is an open access article distributed under the terms of the Creative Commons Attribution License, which permits unrestricted use, distribution, reproduction and adaptation in any medium and for any purpose provided that it is properly attributed. For attribution, the original author(s), title, publication source (PeerJ Computer Science) and either DOI or URL of the article must be cited.
License URL: https://creativecommons.org/licenses/by/4.0/

Keywords: Wireless sensor networks (WSN), Cyclic analysis method (CAM), Forward selection approach, Backward elimination method, CONNECT attack

Funding: The authors received no funding for this work.

==============================
IoT-wireless sensor networks (WSN) have extensive applications in diverse fields such as battlegrounds, commercial sectors, habitat monitoring, buildings, smart homes, and traffic surveillance. WSNs are susceptible to various types of attacks, such as malicious attacks, false data injection attacks, traffic attacks, and HTTP flood attacks. CONNECT attack is a novel attack in WSN. CONNECT attack plays a crucial role through disrupting packet transmission and node connections and significantly impacts CPU performance. Detecting and preventing CONNECT attacks is imperative for enhancing WSN efficiency. During a CONNECT attack, nodes fail to respond to legitimate requests, resulting in connectivity delays, acknowledgment delays, and packet drop attacks in IoT-WSN nodes. This article introduces an Intrusion Detection Algorithm based on the Cyclic Analysis Method (CAM), which incorporates a forward selection approach and backward elimination method. CAM analyzes routing information and behavior within the WSN, facilitating the identification of malicious paths and nodes. The proposed approach aims to pinpoint and mitigate the risks associated with CONNECT attacks, emphasizing the identification of malevolent pathways and nodes while establishing multiple disjoint loop-free routes for seamless data delivery in the IoT-WSN. Furthermore, the performance of CAM is assessed based on metrics such as malicious node detection accuracy, connectivity, packet loss, and network traffic. Simulation results using Matlab software demonstrate superior accuracy in malicious node detection, achieving accuracy in attack detection of approximately 99%, surpassing traditional algorithms accuracy of attack detection.

Introduction

In wireless sensor networks (WSN), the disruption of routing links between nodes can occur as a result of various types of attacks. These attacks encompass a range of threats, including malicious attacks, false data injection attacks, traffic attacks, HTTP flood attacks, non-repudiation attacks, eavesdropping attacks, jamming attacks, clock synchronization attacks, spoofing attacks, node replication attacks, and CONNECT attacks. Notably, the CONNECT attack is distinctive in that it severs links between all nodes along the routing path, whereas other attacks focus on a specific node or route. The CONNECT attack, in particular, indiscriminately severs links between nodes during the routing process, intentionally causing disruption to the connections between nodes and hindering the normal operation of the WSN. Detecting nodes engaged in CONNECT attacks and mitigating their impact during data routing necessitates the implementation of an efficient algorithm. An example of a related threat is the HTTP CONNECT flood, an application layer Distributed Denial of Service (DDoS) attack that specifically targets nodes and their associated applications. The CONNECT Attack is particularly concerning as it disrupts the normal functioning of wireless sensor networks (WSNs) by intentionally severing links between nodes during the routing process. This indiscriminate action hampers the connections between nodes, requiring the implementation of an efficient algorithm to detect and mitigate the impact of nodes engaged in CONNECT attacks during data routing. An analogous threat is the HTTP CONNECT flood, an application layer DDoS attack that specifically targets nodes and their associated applications. Cascading Denial of Service (DoS) attacks further exacerbate the issue by causing congestion throughout the entire network. The attacker manipulates the throughput of a distant node, effectively causing the connection to vanish. Predicting attacks on Wi-Fi networks involves utilizing the phase transition method (Xin, Starobinski & Noubir, 2020). An impersonation attack occurs when an adversary disguises themselves as a legitimate party in a system or communication protocol. To counteract impersonation attacks, deep-feature extraction and selection (D-FES) techniques, combining stacked feature extraction and weighted feature selection, are employed (Aminanto et al., 2018). Recent attacks, including jamming attacks, which target communication protocols like Wi-Fi and Bluetooth, highlight the need for security in wireless communication. The prevention of jamming attacks involves the use of the Moving Target Defense method, which requires further enhancement, especially concerning connection attacks (Alshawi et al., 2020). Wi-Fi networks are susceptible to disruptions in services, interception of sensitive data, and unauthorized access to systems. Predicting the class of attacks on Wi-Fi networks is accomplished using a Support Vector Machine (SVM) (Villain et al., 2019). Services enabled by eIDAS, such as login and Wi-Fi access, are prone to attacks and necessitate robust security algorithms (Berbecaru, Lioy & Cameroni, 2020). In WSN, intrusion detection is imperative due to high data transfer. To address low accuracy resulting from attacks in WSN, the Multiple Convolutional and Gated Recurrent Unit (MC-GRU) method is employed (Jingjing et al., 2022).

The lifecycle of a cyber-attack, often referred to as the “cyber kill chain,” is a concept that outlines the stages an attacker goes through to successfully execute a cyber-attack the typical stages are the reconnaissance, weaponization, delivery, exploitation, installation, command and control, actions on objectives. Moreover, understanding the lifecycle of an attack helps organizations anticipate potential threats and strengthen their defenses accordingly. It is a dynamic process, with attackers constantly evolving their tactics to bypass security measures. Therefore, maintaining vigilance through continuous monitoring, regular training, and updates to security practices is essential for protecting against cyber threats.

In this article, connect attack in the WSN network is shown in Fig. 1 with CONNECT attack occurs through pay load or port scans i.e., IP address in IoT-WSN. CONNECT attack happens through port scans needs to be identified through frequent disconnection in IoT-WSN. Connection attack occurs in two or many nodes in WSN. In this article, connection link breaks through a node in IoT-WSN. For detection and prevention CONNECT attack is performed through CAM method.

Figure 1 Connect attack in WSN.

Image credit: cyber security laboratory at SRM Institute of Science and Technology.

Research problem and research gap

In WSN, the various novel attacks such as DELETE attack, CONNECT attack, OPTIONS attack, TRACE attack need to be addressed in WSN. The traditional algorithms perform IDS for the following attacks such as (i) Sybil attacks, (ii) wormhole attacks, (iii) black hole attacks, (iv) DoS attacks, (v) clone attacks, (vi) replication attacks, (vii) flood attacks.

Problem statement

A node in IoT–WSN passes continuous requests, such as single, double, triple or multiple requests, for attacking other nodes. CONNECT Flood consists of CONNECT requests. When the IoT-WSN node limits concurrent connections. The node responds to legitimate requests from other nodes and attempts to connect, which causes denial of service. CONNECT flood attack detection is quite challenging of differentiation during requests because CONNECT attack node requires is similar to normal nodes. CONNECT attack has more fake connections with the incomplete request generated and sent to the node until the node is overwhelmed with links and starts to stop responding. To solve the above problem, CAM is a method, which iteratively examines the statistical significance of node parameters through a linear regression model and identifies the connect attack nodes. The forward selection approach predicts the connect attack and alternates the routes in IoT-WSN.

Motivation

CONNECT attackers attack multiple systems in the network simultaneously because the attack happens during the crashing of website link and breaking connection links, between website and the user, which happen in few second and within that few seconds data are stolen. Moreover, during the short span of connection failure, connect attack is performed. The CAM method is proposed to detect the CONNECT attack in IoT-WSN. The CONNECT attack occurs during the short span of time, which is detected through proposed CAM method. Until now, attack detection in the Adhoc network, WSN and sensor are performed for various attack as mentioned above. The IoT-WSN based cyber-physical system attacks need analysed and detection of attacker nodes in IoT-WSN needs efficient algorithm.

Contributions

1) To propose CAM method with minimal time complexity for Intrusion Detection Systems (IDS) for CONNECT attack in an IoT-WSN environment, and detection ratio is introduced for rapid detection of attack.

2) To detect connect attack at different scenarios, which causes the loss of connectivity is identified through WSN traffic, and CPU utilization time and memory usage dataset of collected from nodes in IoT-WSN.

3) To compare the proposed CAM, with traditional method during disjoint loop-free routes in WSN.

Literature works

Advancements in embedded systems and the Internet of Things (IoT) have given rise to an increased frequency of attacks in WSN. Within the realm of WSN, various conventional attack types pose significant threats, including malicious attacks, false data injection attacks, traffic attacks, HTTP flood attacks, non-repudiation attacks, and eavesdropping attacks. Malicious attacks stem from data sensors in IoT nodes and can be thwarted by lightweight technologies employing distance measurement (DM) algorithms. This approach is exemplified by the implementation of the CERT (Community Emergency Response Team) dataset (Khan et al., 2019). Detecting false data injection attacks in smart meters involves the use of deep learning, machine learning, and parallel computing algorithms. Cyber, physical, and social interfaces make smart meters susceptible to data manipulations in real-time data collected from the smart grid (Unal et al., 2021). To mitigate Distributed Denial-of-Service (DDoS) attacks, a signature-based traffic classification system is employed. Unique packet feature combinations are identified, and machine learning algorithms are used to counteract DDoS attacks. The efficacy of this signature-based protection algorithm is assessed using Booster datasets (Dimolianis, Pavlidis & Maglaris, 2021). Eavesdropping attacks, conducted through UAV system sneaking during processing, are addressed by analyzing datasets prepared using wireless signals. The analysis involves the application of the proposed one-class support vector machines (OC-SVM) and K-means clustering (Hoang, Nguyen & Duong, 2019). Interest flooding attacks (IFA) are mitigated through hypothesis testing theory and likelihood ratio tests using datasets from the Montimage Monitoring Tool (MMT) Probe (Nguyen et al., 2019). In 5G Cloud Radio Access Networks (C-RAN), jamming attacks are detected and classified through a combination of deep learning and kernelized support vector techniques. The Wireless Sensor Network dataset (WSN-DS) serves as the testing ground for jamming attack detection (Hachimi et al., 2020). Clock synchronization attacks on electrical grids, specifically targeting servo clocks in the IEEE bus bar system, are addressed by mitigating the attack in the 39-bus IEEE benchmark dataset. This mitigation involves measuring two distinct phases simultaneously (Shereen et al., 2020). Node replication attacks in industrial WSNs, occurring at the physical layer, are resolved using the Secure Random Key Distribution (SRKD) method. SRKD is a hybridization of localized algorithms and voting methods for the detection and revocation of malicious nodes (Li et al., 2019). In Bluetooth low energy (BLE), wireless sniffing attacks leading to jamming, encryption cracking, or system penetration are countered through the BLE channel selection algorithm. The algorithm’s performance is evaluated using datasets collected from the Wireshark network monitoring tool (Sarkar, Liu & Jovanov, 2019). Denial-of-Service Attacks in Wi-Fi Direct Device-to-Device Networks are thoroughly studied and analyzed. Such attacks force the Wi-Fi connection to drop and prevent the detection of the access point or cellular network (Sankaranarayanan, Gayathri & Tamijetchelvy, 2021; Hadiks et al., 2014). Wi-Fi networks are susceptible to impersonation attacks, where SSID or MAC/IP addresses mimic those of legitimate devices (Liu et al., 2019). Additionally, the security compromise resulting from Evil Twin attacks (ETA) on connected devices is addressed within the Wi-Fi context (Kuo, Chang & Kao, 2018). The Table 1 shows the types of attack.

Table 1 Types of attack.

(Ref)/(Year)	Different attack	Network device	Prevention	Remarks	
De Araujo-Filho et al. (2021)	Hindering cyber-attacks	CAN	Unsupervised intrusion prevention system	Improves security	
Mwinuka et al. (2022)	Network spoofing attack	WSN	Android-based client-side	Fake access is prevented	
Huang et al. (2022)	forgery attacks	WSN	Wireless signals to detect looped videos	Surveillance systems	
Tran Le et al. (2022)	Malware attacks	WSN networks	Epidemiological mathematical model	Encryption and authentication in Wi-Fi	
Li et al. (2021)	impersonation attack	Wi-Fi	Meta-WF	AWID dataset	
Sharma, Elmiligi & Gebali (2021)	Hello flood attack, increased version attack, decreased rank attack, DIS attack.	Wireless sensor network.	SVM, Bayes classifier, random forest.	Reinforcement learning needs to be used.	
Sun, Yuan & Fan (2022)	Real world attack connections	WSN	High–interaction honeypot (HIH), Low–interaction honeypot (LIH)	Time related connection properties dynamically need to be used	
Kaya & Selcuk (2020)	Post connection attacks	Wireless sensor network	Automated database scanning method	Analysis of security is need to be improved	
Lounis & Zulkernine (2020)	WPA3 connection attacks	WSN	WPA3-SAE (Simultaneous Authentication of Equals)	Improve security	
Salmi & Oughdir (2022)	DoS attacks	WSN	CNN–LSTM method	Improves network security	
Proposed	DELETE attack, CONNECT attack, OPTIONS attack and TRACE attack	WSN	Cyclic analysis method (CAM)	Reduces time complexity and space complexity	

Inferences from literature survey

The unsupervised learning algorithm performs less due to unlabeled data. The increase in over fit risk occurs when the number of observations are insufficient during execution of IDS (De Araujo-Filho et al., 2021; Mwinuka et al., 2022). Existing IDS algorithms, needs more computation time when the number of variables is more. Moving target defense never prevents the attacker from the building a weaponized attack. The support vector machine does not perform for large dataset because the required training time is higher. The naive Bayes performs based on independent predictor features. It is to get a set of predictors that are completely independent (Huang et al., 2022; Tran Le et al., 2022; Li et al., 2021). The random forest has large no of trees and ineffective for real time predictions (Sharma, Elmiligi & Gebali, 2021; Sun, Yuan & Fan, 2022).

Materials and method

In IoT-WSN, CONNECT attack performs through a false message through the malicious node and misguides the node (Lounis & Zulkernine, 2020). In Fig. 2, dotted lines are on the wrong path, which is due to malicious nodes, which performs CONNECT attack. The malicious route looks like the shortest route in IoT-WSN. The CONNECT attack targets the particular node through IP address and port address (Salmi & Oughdir, 2022). IP address and port address are misused. The receiver could not reach the proper data from the source address in IoT-WSN when the CONNECT attack happens in the particular node at a scheduled time to hack the IP address and creates a false route for data transfer (Ramat Gan, 2000). In turn, target node reduces the CPU performance. At the same time, the packet drop or delay, acknowledgment delay and loss of connectivity are occurred. The CONNECT attack is detected by using Cyclic Analysis Method (CAM). In CAM, the nodes CPU performance data are collected and analyzed. Data from each node CPU is collected and termed as CA-dataset. CA-dataset consists of the following data parameters values such as CPU performance, Memory usage, RAM usage, disk usage, Wi-Fi send packets, Wi-Fi received packets, GPU usage, and active time.

Figure 2 CONNECT attack architecture diagram.

Image credit: Cyber Security Laboratory at SRM Institute of Science and Technology.

Test bed for CONNECT attack

Raspberry Pi-based attack detection testbed involves setting up a network environment for simulation and detection of various cyber-attacks. This setup is configured with 4 Raspberry Pi and security tool to monitor and analyze network traffic for malicious activities or policy violations. Raspberry Pi 4 model is suitable due to improved CPU speed, memory, and network capabilities compared to older models. Raspberry Pi OS-kali Linux and MATLAB are installed, then updated and upgraded it to the latest versions of the software through bash commands such as sudo apt-get update and sudo apt-get upgrade. Further, installed attack detection tools were used for data collection.

Dataset description

The CONNECT attack targets the IP and port address. It shows the local host address and connects the remote host address and port address by injecting the CONNECT attack through the node. The TCP protocol is used to attack the target node in IoT-WSN. After the CONNECT attack happens, it shows the remote hostname and port number. CPU performance asses through clock speed, i.e., clock rate - indicates CPU speed is measured as megahertz (MHz) or gigahertz (GHz), which deals with how the number of instruction cycles the CPU delt in seconds. A 2 GHz CPU deals with two billion cycles/second. Clock speed counts the number of cycles in GHz (gigahertz). Memory is continuously updated, and RAM is used by each process at the given moment. Disk usage is the percentage of hard disk currently being used by the microcontroller to run programs and carry out tasks. Wi-Fi send packets are sent via radio waves. Wi-Fi received packet means that every webpage receive a series of packets. GPU Usage is for intensive tasks. Active time reflects the time span from the network’s initial deployment to the first loss of coverage. Network lifetime is defined as the time until the first node dies.

With a source dataset of 1 MB and a sample size of 16,000, we can run 75 training runs and 25 test runs. A test WSN was set up and evaluated, and based on that experience, an experimental WSN was deployed in SRM TECH PARK. Thirty sensor nodes were linked to thirty systems and instructed to transmit a data payload of fifty-seven bytes every 1 s. An interface shows the received data from the base station to the PC in the control room. The gathered data is recorded by providing a dataset to the GUI. An analysis was conducted on the data obtained from 30 sensor nodes over the course of 1 h. At 1 s intervals, thirty sensor nodes were sending data to the base station. Every node sent 3,600 packets in 1 h. The experimental network with 30 nodes produced 1,08,000 data packets in 1 h. The database revealed that the base station received 1,08,000 packets per hour. Therefore, we saw a 100% delivery rate for the packets. CAM for Connect Attack Mitigation in WSN

The CAM consists of three phases such as recognition of suspicious nodes, identification of CONNECT attack nodes and elimination of the CONNECT attack nodes. To identify the suspicious nodes, two connected routing paths, such as shorter and longer connected routing paths. The shorter connected path is connected to the sink node, such as the connected routing path A -> CA -> SN, as in Fig. 3. The longer connected path is a routing path, where they have at least four sensor nodes from the event node to the neighbor node, such as two types of connected routing paths A -> C -> E -> SN and A -> B -> D -> F -> SN. On a shorter connected routing path, the intermediate node is the suspicious nodes. In a longer connected routing path, the neighborhood information is used for the recognition of the suspicious nodes. For identification of CONNECT attack nodes, the number of interaction times and ACK are used to judge suspicious nodes, which are CONNECT attack nodes. To elimination of the CONNECT Attack nodes, the event node removes CONNECT Attack node from the network. The ID information of CONNECT attack nodes is removed from the routing table.

Figure 3 Two types of connected routers.

In this article, CAM routing protocol is proposed for detection of CONNECT attacks, as shown in Fig. 4. Node A is not in range of the sink node and cannot communicate directly, and it can exchange packets via routing through nodes C and E or by node B, node D and node F. When node A sends packets to the sink node (SN), it initially broadcasts packets in one–hop to the next nodes, for example, node B and node C. In this case, CONNECT attack node looks at the reply packets, changes the hop count by itself and sends to the sink node and fabricates a short route to the sink node. Modified hop count is included in reply packets and sent back to node A. When node A receives reply packets from the neighbor node and the CONNECT attack node. The route through the CONNECT attack node is the shorter one and sends packets to the sink node through CONNECT attack node.

Figure 4 Example of CONNECT attack.

Advantages and limitations of CAM

CAM is used in various fields, such as engineering, and technical analysis of financial markets, analyzes the repeating patterns or cycles in data. The key advantages of using the CAM are the (i) predictive power i.e., cyclic analysis allows for the prediction of future events or trends based on identified patterns.(ii) Improves understanding of systems through deeper understanding of the systems being studied by highlighting recurring events or patterns. (iii) Complementarity, which is done based on the conjunction with other analytical methods to enhance the overall analysis. For example, in financial markets, cyclic analysis might be used alongside fundamental and technical analysis to provide a more comprehensive view. (iv) Enhanced decision making through a framework for anticipating future events, cyclic analysis helps individuals and organizations make more informed decisions, potentially leading to better outcomes. Moreover, Limitations are the assumption that patterns will repeat exactly or that all phenomena are cyclic can lead to misinterpretations.

Algorithm for the CAM

In IoT-WSN, CONNECT attack nodes have more energy and utilizes the high CPU time and more memory usage than other nodes in the network. CONNECT attack nodes directly communicate with the base station/sink node. During the shortest connecting path in the WSN, among three sensor nodes, the initial node is the event node, the next is an intermediate node, and the final node is the destination node. All nodes have active time (T). The event node broadcasts packets to the destination node within a single-hop communication. Each packet has an ID for the next hop node. Neighbor nodes receive the packet and reply to the event node. When reply packets are received, the event node checks the reply packets. If the next hop node is an intermediate node, then the next is the sink node, whereas the intermediate node is the suspicious node. The details are presented in CAM Algorithm 1. Node A is the event node, and T is the active time.

Algorithm 1 Cyclic Analysis Method (CAM) algorithm.

Forward stepwise selection Algorithm	
Input: CA Dataset, TA Target	
Output: SV Selected Variables	
1: SV ← φ // Set of selected variables	
2: RV ← V // Set of remaining candidate variables	
3:	
4: // Forward Selection: Repeat until SV does not change	
5: while SV changes, V	
6:    // Identity the finest variable V best out of all residual variable RV, affording to PERF	
7: Vbest← argmax PERF (SV V)	
       V € RV	
8:  // Select V best if it increases performance according to condition C	
9:  if PERF (SV Vbest) PERF (SV) then	
10:   SV ← SV V best	
11:   RV ← RV\V best	
12: end if	
13: end while	
14:	
15: // Backward selection: Repeat until SV does not change	
16: while SV changes do	
17:     // Identify the worst variable V worst out of all selected variables SV, affording to PERF	
18:    Vworst← argmax PERF (SV\V)	
            V € SV	
19:    // Remove V worst if it does not provide shrinkage performance according to criteria C	
20:    if PERF (SV < V worst) PERF (SV) then	
21:    SV ← SV\V worst	
22:    endif	
23: end while	
24: return SV	

In this Algorithm 1 is used to denote the following: CA, the dataset, TA, the output and selected variable (the node), and RV, the candidate variable (the node nearest to the selected one). Criteria C for selection uses PERF to compare two sets of performance metrics. The PERF function returns the results of evaluating a set of variables. Starting with a (typically empty) set of variables, forward selection continues to add variables until a stopping criterion is satisfied. Again, until a stopping criterion is satisfied, backward selection begins with a (typically full) set of variables and continues to exclude variables from that set. In most cases, the goal of both approaches is to either incorporate or omit the variable that yields the greatest improvement in performance. Forward (backward) iteration describes each step of adding or removing a variable. One phase is the execution of forward iterations, while the other is the execution of backward iterations until termination. The Forward-Backward Selection algorithm (FBS) is an example of a stepwise algorithm, which will be the main focus of this article. It iteratively applies a forward and backward phase to the variables that have been chosen. To compare two sets of variables, we will use the predicates > C, ⋥ C, and = C. These statements are true when the value on the left-hand side is greater than, greater than or equal to, or equal to the value on the right-hand side, respectively, as determined by the criterion C.

The CONNECT attack node changes the hopping data and performs forwarding attack. CONNECT attack node disrupts traffic through malicious attacks and changes the parameters such as CPU utilization time, memory usage, loss of connectivity and WSN traffic. The CAM detects false data packets.

Forward selection CAM

In the forward selection method, each node data is added during each and every time of regression modeling. This method is used for larger predictor variables. The steps to perform the forward selection are as below. Figure S1A to Table 2 (l) shows the forward selection CAM. The nodes to be connected from one node to another node. At that time, collects the data parameters from the node in IoT-WSN.

Table 2 Comparison of precision, recall, F-measure and accuracy of traditional algorithms vs proposed CAM.

Measure	Precision	Recall	F-measure	Accuracy	Detection ratio	Time complexity	
MC–GRU (Jingjing et al., 2022)	96.93	95.72	96.35	96.86	96.18	0.25	
BLE (Hachimi et al., 2020)	88.17	84.09	90.57	80	84.49	11.82	
OCSVM and iForest (Shereen et al., 2020)	79.52	70.68	74.42	83.58	80.37	11.7	
SVM & NN (Hadiks et al., 2014)	87	86.26	86.98	84.24	86.5	11.83	
Fake AP Detector (Liu et al., 2019)	86.5	80.56	84.9	85.2	94.53	12.2	
Wi-Fi CSI (Kuo, Chang & Kao, 2018)	90.5	93.9	96.12	96.5	92.5	4.83	
DNN (Mwinuka et al., 2022)	93.32	92.47	92.54	95.97	90.46	5.44	
Fuzzy Logic (Kaya & Selcuk, 2020)	75.12	84.09	76.94	82.37	93.36	0.38	
KNN (Lounis & Zulkernine, 2020)	94.96	92.7	96.79	89.97	95.69	0.35	
CNN-LSTM (Ramat Gan, 2000)	95.87	94.96	96.85	96.98	95.24	0.28	
Proposed CAM	98.85	98.97	98.87	99.76	99.83	0.12	

1) Train the ‘n’ model using each feature (n) individually and check the performance.

2) Choose the variable which gives the best performance.

3) Repeat the process and add one variable at a time.

4) Variable with the greatest improvement is retained.

Repeat the entire process until there is no significant improvement in the model’s performance

Backward elimination CAM

In backward (step-down) elimination, where one variable is deleted at a time during the regression model’s progress. Figures S2A–S2H shows the backward elimination CAM. The nodes to be deleted from one node to another node. At that time, collects the data parameters from the node in IoT-WSN. (i) Calculating the t-statistic for the estimated coefficient of every variable in the model.

(ii) Squaring the t-statistic to create the F- to remove the statistic.

Sensitivity analysis of CONNET attack detection using CAM

The algorithms such as CAM, Fuzzy and KNN algorithms are performed in the Raspberry Pi module as nodes IoT–WSN. This proposed framework performs realistic data communication protocols in WSN through NS2 and Data analysis. Figure 5 shows the CPU usage before CONNECT attack. The blue dot represents CPU usage. The orange dot represents the Memory usage of CONNECT attack. The data point blue and orange dots represented more distance during the normal routine. The arrow mark represents the distance of routing. Figure 6 shows the CPU usage after CONNECT attack. The blue dot represents CPU usage. The orange dot represents the Memory usage of CONNECT attack. The data point’s blue and orange dots are represented as the short distance during the CONNECT attack routing. The arrow mark represents the distance of CONNECT attack routing. Figure 7 shows the connectivity loss of Fuzzy logic, KNN and CAM. Compared to the traditional model, CAM gives less connectivity loss (5%) of nodes in IoT–WSN. Figure 8 shows subtracting the delays of two successive requests in node links. The reliability of a communication network path is represented as packet loss rate (3%) in IoT – WSN. This metric is equal to the number of packets not received divided by the total number of packets sent in WSN. Compared to Fuzzy logic, KNN and CAM give a lower packet loss rate in WSN. Figure 9 shows the accuracy (99%) of CONNECT attack in WSN. Using CAM algorithm gives better accuracy of WSN compared to Fuzzy logic and KNN. Figure 10 shows the Acknowledgement delay of CONNECT attack in IoT–WSN. The source node sends the ACK to the receiver node received. The value of the delayed ACK and the active timer is irrelevant in WSN. Figure 10 shows less acknowledgment delay (2%) in IoT–WSN compared to other traditional fuzzy logic and KNN. Table 2 shows comparison of precision, recall, F-measure and accuracy of traditional algorithms Vs proposed CAM.

Figure 5 CPU usage vs memory usage before CONNECT attack.

Figure 6 CPU usage vs memory usage after CONNECT attack.

Figure 7 Connectivity loss.

Figure 8 Packet loss rate.

Figure 9 Accuracy.

Figure 10 Acknowledgment delay.

Figures 11–16 shows the precision, recall, F-measure, accuracy, detection ratio and minimal time complexity of CONNECT attack compared with traditional algorithms.

Figure 11 Comparison of precision for traditiona algorithms and CAM for CONNECT attack.

Figure 12 Comparison of recall for traditional algorithms and CAM for CONNECT attack.

Figure 13 Comparison of F-measure for traditional algorithms and CAM for CONNECT attack.

Figure 14 Comparison of accuracy for traditional algorithms.

Figure 15 Comparison of accuracy for traditional algorithms and CAM for CONNECT attack.

Figure 16 Comparison of detection ratio for traditional algorithms and CAM for CONNECT attack.

Conclusion

CONNECT attacks are frequent in IoT- WSN. CONNECT attacks occurs in WSN, which discards the packet and node connection of WSN and hard to detect the node performs CONNECT attack. The CAM is proposed, which consists of two process, initially forward selection approach and second backward elimination method. The CAM generates routing information and analysis behavior of IoT-WSN for detecting CONNECT attack and finds the multiple disjoint loop-free routes in IoT-WSN. CAM acquires the data parameters such as CPU usage, Memory usage from each node and applies forward and backward selection based model for detection of CONNECT attack nodes in IoT-WSN. CAM performance is analyzed through CONNECT attack node detection accuracy, connectivity, packet loss and network traffic. CAM accuracy in CONNECT attack node detects of about 99% than traditional algorithms. CAM detects CONNECT attack nodes and efficiently identifies all the CONNECT attacked nodes. The advantages of CAM method in detecting the CONNECT attack are less time consumption and no need for hardware devices. CAM is a lightweight algorithm and detects the CONNECT attack through the node’s data. From the experimental results, CAM algorithm can be further improved by deep learning hybrid algorithm for heterogeneous nodes in the network.

Supplemental Information

Supplemental Information 1 Forward Selection CAM.

Supplemental Information 2 Backward Elimination CAM.

Supplemental Information 3 Connect attack Dataset.

Supplemental Information 4 EternalBlue is a serious cyber vulnerability discovered in Microsoft Windows operating systems.

In particular, it uses a flaw in the Server Message Block (SMB) protocol seen in versions of the protocol older than Windows 8 and Windows Server 2012. This hack primarily exploits the MS17-010 security hole, allowing malware to spread across networks at alarming speed. EternalBlue emphasizes the need for timely software upgrades and strong cyber security procedures to reduce risks from such severe vulnerabilities.

Supplemental Information 5 Specifically aimed at the SMB protocol, this Python software makes use of a flaw in Microsoft Windows systems using the struct module and impact library to cause arbitrary code execution and a buffer overflow.

The script gains control of the machine by exploiting the SMB protocol's buffer overflow vulnerability.

Systems Targeted: Specifically, Windows 2008 and Windows 7 SP1 systems are targeted.

Payload Construction: To exploit the issue, the script builds payloads with exact buffer sizes of 0x10000 and 0x11000 bytes.

Memory Manipulation: To enable unrestricted write and code execution, the script modifies memory structures such as MDLs (Memory Descriptor Lists) and buffers in srvnet.sys.

Shell code Execution: The exploit makes it possible for shell code to run in kernel mode, which opens the door for malware to install or other harmful operations.

Additional Information and Declarations

Competing Interests

Author Contributions

Data Availability

The authors declare that they have no competing interests.

Tamil Selvi S conceived and designed the experiments, performed the experiments, analyzed the data, performed the computation work, prepared figures and/or tables, authored or reviewed drafts of the article, and approved the final draft.

Visalakshi P analyzed the data, performed the computation work, authored or reviewed drafts of the article, and approved the final draft.

The following information was supplied regarding data availability:

The raw data and code are available in Supplemental Files.

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
