# Peer review of "Connect attack in IoT-WSN detect through cyclic analysis based on forward and backward elimination"

_PeerJ Computer Science, doi:10.7717/peerj-cs.2130_

## Round 0.1 · original submission · Major Revisions

The manuscript has several weaknesses, including the lack of clear illustrations for the proposed method's three steps, an undefined attack model for CONNECT attacks, insufficient details about the experimental setup and methodology, and a limited comparison with state-of-the-art approaches. Additionally, the paper's overall innovation in addressing CONNECT attacks is unclear. Addressing these issues is crucial to improving the manuscript's clarity, experimental validity, and overall readiness for publication.

**Language Note:** PeerJ staff have identified that the English language needs to be improved. When you prepare your next revision, please either (i) have a colleague who is proficient in English and familiar with the subject matter review your manuscript, or (ii) contact a professional editing service to review your manuscript. PeerJ can provide language editing services - you can contact us at [email protected] for pricing (be sure to provide your manuscript number and title). – PeerJ Staff

Reviewer 1 ·

Basic reporting

This paper introduces an Intrusion Detection Algorithm utilizing the Cyclic Analysis Method (CAM) to combat CONNECT attacks in Wireless Sensor Networks (WSNs). The proposed algorithm outperforms traditional methods, as evidenced by simulation results in Matlab. Despite its promising outcomes, the paper would benefit from a more comprehensive comparison with state-of-the-art approaches. The clarity and focus on the unique features and advantages of the proposed method could be improved in the introduction. Addressing these aspects would enhance the overall impact and contribution of the paper in the field of WSN security.

Experimental design

The experimental setup lacks clarity and fails to provide a comparison with state-of-the-art methods.

Validity of the findings

The experimental design and algorithm description lack clarity, affecting the study's replicability. Inadequate details about data and algorithm steps, coupled with a limited comparison with existing methods, create uncertainty. Moreover, the proposed method, while addressing the identified issue, lacks significant innovation compared to current approaches. Improved transparency in experiments and algorithmic steps is essential to ensure the reliability and applicability of the reported findings.

Additional comments

1. The introduction contains excessive information on attacks and defense mechanisms unrelated to the paper's focus. It is essential to highlight the unique features and advantages of the proposed approach. Additionally, a section on related work specifically addressing CONNECT attack defense is recommended.
2. Section 2.a's data description lacks details on quantity, types, and sizes of data (both training and testing). The author should provide this information rather than relying solely on various names and explanations.
3. There is a mathematical symbol mix-up in Algorithm 1.
4. The description of the CAM algorithm is unclear. For instance, in Algorithm 1, what does PERF refer to? What features are considered in section 2.d, and what exactly does the term "model" refer to? Additionally, the assumption of having prior attack data for identifying attackers might be too strong due to the difficulty and cost of collecting comprehensive attacker information.
5. In the experiments, the analysis of the proposed method's performance under various conditions should be presented separately. The comparison with older methods lacks depth, and advanced methods are not considered. Moreover, there is no evaluation of time efficiency and data cost efficiency (i.e., how much training data is required). The metrics and settings of the experiments are not well-defined.
6. The quality of the images is subpar, with deformation and small lettering. Figure 5 lacks proper labeling, making it unclear which data represent the effects of the CAM method.

**Typo:**
1. On line 83, IDS is first mentioned, not on line 111.

Reviewer 2 ·

Basic reporting

This paper proposes a new method to detect the CONNECT attacks in Wireless Sensor Networks (WSNs).
Although the problem seems important, the current state of the paper is not ready for publication.

The language of this paper is fine, but the illustration of the proposed method is substantially lacking.
The proposed method consists of three steps.
For the first step, (Section 2c), the algorithm 1 has typesetting issues and is not readable.
The other two steps (Section 2d and 2e) only have several bullet points without any illustration.
Based on the current state of the description, it is hard to understand the proposed method.

Before the literature review, the authors do not clearly state the attack model for the CONNECT attack.
For example, in Section 1.2, the authors only high-levelly describe the importance
and potential consequences of the CONNECT attack, but do not clearly state
what the CONNECT attack is and what the attack model is.
Specifically, the authors should clearly state the following:
1. The attack has access to which part of the network? and has which capabilities?
2. What is the "lifecycle" of the attack? (e.g., is the attack consists of multiple steps?)
3. What is the goal of the attack? I understand that in general DoS attack aims
to make the service unavailable. However, the authors mention "data theft" at line 105.
Is the goal of the CONNECT attack to steal data? and how?

For the experiments, the authors use figures to show that
the proposed method has a better precision, and reduces the packet loss rate.
However, there is no details about the experimental setup (e.g., how CPU usages, package loss rates are computed).

Experimental design

The authors use several figures to illustrate the system performance without any details about experimental setup. From the paper, it is not sufficient to tell whether the experiment is meaningful.

Validity of the findings

The authors do not provide enough details about the system design and the experiments.
It is hard to tell whether the findings are valid.

---

## Round 0.2 · Minor Revisions

The revised version of the paper adequately addresses the majority of the concerns raised by the reviewer, with improvements made to focus the abstract and introduction on CONNECT attacks. Efforts have been made to enhance the clarity of Algorithm 1 through additional illustrations, though suggestions regarding font choice could further improve readability. While descriptions regarding the network access and capabilities of the attack have been added, further clarification on the attack's lifecycle and goals, as discussed in the rebuttal, should be incorporated into the paper. Formatting issues, particularly with section titles and symbols, still require attention. The description of experimental evaluations has been refined, but more depth is needed to delineate individual experiments and sub-experiments. Additionally, a more comprehensive analysis of limitations and elucidation of specific advantages over traditional CAN algorithms would enhance the paper's contribution. Although the majority of concerns have been addressed, some clarifications provided in the rebuttal have not been reflected in the revised paper, and formatting issues persist, requiring resolution for improved manuscript quality and clarity.

Reviewer 1 ·

Basic reporting

i) The abstract and introduction provide an overview of various attacks, some of which are not directly relevant to the study's focus. It is advisable for the author to streamline the discussion to concentrate solely on CONNECT attacks, elucidating their significance, the proposed methodology, and the comparative advantages over traditional approaches.
ii) The article exhibits inconsistencies in paragraph indentation and formatting, necessitating corrections for uniformity.
iii) There is room for improvement in the author's English writing proficiency.

Experimental design

The description of experimental evaluations lacks depth, particularly in delineating individual experiments and sub-experiments. A more detailed discussion is warranted to provide clarity and context.

Validity of the findings

i) The paper would benefit from a comprehensive analysis of its limitations, providing readers with a clearer understanding of the scope and applicability of the proposed methodology.
ii) It is essential to elucidate the specific advantages of the author's algorithm over traditional CAN algorithms, enhancing the paper's contribution to the field.

Additional comments

The formatting of section 1.4's title needs correction. There appears to be a formatting error involving symbols on line 254 that requires attention.

Reviewer 2 ·

Basic reporting

I am fine with the new version. Overall, I think the authors address most of my concerns in the response or the revised paper.
The authors clarify most of my concerns in the response,
although some clarifications are not reflected in the revised paper.

I had the following concerns in my previous review:

1. Algorithm 1 is hard to understand

The authors address this concern by adding more illustrations after Algorithm 1. Now it becomes clearer. Minor: Maybe is a better idea to use more easy to read fonts for the algorithm (e.g., maybe consider using normal serif fonts or using tt fonts).

2. The attack has access to which part of the network? and has which capabilities?

The authors clarify this question by adding descriptions in the introduction.

3. What is the "lifecycle" of the attack? (e.g., is the attack consists of multiple steps?)

The authors explain the lifecycle of the attack in detail in the rebuttal text. However, the explanations are not added to the paper.

4. What is the goal of the attack?

The authors clarify the scope of the attack in rebuttal. CONNECT attack aims to block the webpage request, but not to steal data.

5. The detailed experimental setups.

The authors detailed discuss the experimental setups both in rebuttal and the revised paper. Also, the authors clarified what are the commands used to monitor system status.

Experimental design

no more comments

Validity of the findings

no more comments

Additional comments

no more comments

---

## Round 0.3 · accepted · Accept

Thank you for the significant improvements made to your manuscript. I am pleased to recommend your paper for acceptance.

However, please address the following to enhance presentation quality: Ensure the font style and line spacing for the algorithm match the rest of the paper and check for formatting errors in the PDF. Maintain consistency in paragraph indentation throughout the paper.

Reviewer 1 ·

Basic reporting

Thank you for the improvements made to your manuscript. I would like to provide some suggestions to further refine the presentation:
1. The formatting of the bold font and line spacing for the algorithm is not consistent with the overall style of the paper. I recommend selecting a font style that aligns with the rest of the document and ensuring that there are no formatting errors, particularly after the manuscript is exported to PDF format.
2. There appears to be inconsistency in paragraph indentation throughout the paper. Some paragraphs are indented, while others are not. It would be beneficial to carefully review and address this issue to maintain uniformity in the document's formatting.
I trust that these suggestions will enhance the overall readability and presentation quality of your manuscript.

Experimental design

N/A

Validity of the findings

N/A

Additional comments

N/A